# Evaluation of Reliquefaction Behavior of Coastal Embankment Due to Successive Earthquakes Based on Shaking Table Tests

**Mintaek Yoo** [1] and **Sun Yong Kwon** [2,*]

1    Department of Civil & Environmental Eng., Gachon University, 1342 Seongnam-daero,
     Seongnam-si 13120, Republic of Korea; mintaekyoo@gachon.ac.kr
2    Environmental Assessment Group, Korea Environment Institute, 370 Sicheong-daero,
     Sejong-si 30147, Republic of Korea
*    Correspondence: sykwon@kei.re.kr

**Abstract:** Liquefaction caused by long-term cyclic loads in loose saturated soil can lead to ground subsidence and superstructure failures. To address this issue, this study aimed to emulate the liquefaction phenomenon based on a shaking table test while especially focusing on the soil behavior mechanism due to the reliquefaction effect. Liquefaction and reliquefaction behaviors were analyzed by ground conditions where an embankment was located on the coastal ground. Silica sand was used for the experiment for various thickness and liquefiable conditions, and the embankment model was constructed above the model ground. For seismic waves, sine wave excitation was applied, and a total of five excitations (cases) were conducted. When the upper ground layer consisted of a non-liquefiable layer, liquefaction did not occur due to the first excitations but occurred by the third excitation. The results indicated that as the earthquake was applied, the water level in the liquefiable layer increased to the height of the non-liquefiable layer and liquefaction could occur. It was identified that even if liquefaction did not occur for the main earthquake, liquefaction could occur due to aftershocks caused by a rise in the groundwater level due to a series of earthquakes. In a general seismic design code, liquefaction assessment is performed only for soil layers below the groundwater level; however, when successive earthquakes occur, unexpected liquefaction damage could occur. Therefore, to mitigate the earthquake risk of liquefaction for coastal embankments, it is necessary to evaluate the liquefaction by aftershocks even when the groundwater level of the ground layer under an embankment is low.

**Keywords:** reliquefaction; liquefaction; coastal embankment; excess pore pressure; aftershock

## 1. Introduction

In recent years, the coast has been reclaimed in several areas of the world for the development of industrial complexes, wind power generation, tourism complexes, etc., and coastal areas with high liquefaction concerns are increasing. In particular, with the increasing number of cases of constructing a coastal embankment after reclamation and using the embankment as a foundation for a wind power generation facility or using it as a walking trail or bicycle road for tourism effects, concerns over liquefaction damage to coastal embankments are growing. Liquefaction is a phenomenon in which soil loses its resistance and behaves in a manner similar to a liquid due to a gradual increase in excess pore water pressure caused by long-term cyclic loads in loose saturated soil. When liquefaction occurs, the soil loses its strength and ability to support superstructures such as buildings and bridges, which can lead to ground subsidence and superstructure failures.

On the other hand, soil densification due to rearrangement and reconsolidation of particles after liquefaction increases the resistance in future earthquakes, and this theoretical mechanism is widely applied to ground improvement construction such as compaction to prevent liquefaction. However, some cases suggest that this intuitive theory is not

necessarily applicable in all cases. After the first liquefaction, severe cases of reliquefaction due to aftershocks have been continuously reported [1–6], and the results of related lab experiments also support these cases [7–11]. To understand and prepare for the dynamic behavior of the ground that is different from the widely known general theory, in-depth research on aftershock and reliquefaction mechanisms is required, but related research is still lacking. Moreover, the recent Türkiye–Syria earthquake resulted in more serious damage due to aftershocks than due to the main earthquake, reminding us of the need for research on aftershocks, which have been relatively understudied.

Based on the 1983 Nihonkai–Chubu earthquake, Ohara et al. found that liquefaction occurs in the soil at lower peak ground acceleration and shear stress ratio values than those at the initial states, and reliquefaction can occur during earthquakes with magnitudes smaller than those of earthquakes that occurred previously [12]. Oda et al. investigated the reliquefaction behavior of saturated granular soils through lab tests [13]. They identified that liquefaction resistance was significantly lost if large excess pore pressure was generated in the first cycle. Furthermore, they investigated the influence of several soil parameters such as the void ratio, relative density, inherent isotropy, and void shape on reliquefaction behavior. Zhao and Ye performed a series of 3D DEM simulations of undrained cyclic triaxial tests, and the entire process of main shock-induced liquefaction, reconsolidation with various degrees, and aftershock-induced reliquefaction was reproduced [14]. They identified that the reliquefaction resistance of a completely reconsolidated soil may be higher or lower than its initial liquefaction resistance, which is mainly affected by the residual anisotropy caused by the first liquefaction. They also identified that reliquefaction resistance is significantly affected by the reconsolidation degree. By employing a shaking table test, Ha et al. created soils based on five types of sand tests with a high likelihood of liquefaction and analyzed the effects of the sand gradation characteristics on changes in the reliquefaction resistance during reliquefaction [15]. They observed that as $D_{10}/C_u$ increased, the reduction rate of the reliquefaction resistance decreased linearly, and that as $D_{10}/C_u$ exceeded 0.15 mm, the reduction rate of resistance during reliquefaction was constant at approximately 20%. Moreover, based on the number of loading cycles and excess pore water pressure over time, it was confirmed that the probability of liquefaction decreased as the excess pore water pressure decreased when the fourth and fifth shaking loads were applied. In a recent study on reliquefaction, Nepal et al. measured the excess pore water pressure and acceleration response based on reliquefaction experiments [16]. As observed, (a) the liquefaction resistance of sand was lower in the second liquefaction than that in the first, (b) the liquefaction resistance varied with depth, and (c) the probability of liquefaction of the soil layer near the surface was high. In addition, several researchers confirmed that seismic shear wave damping in a liquefiable soil layer had different characteristics from that of a general soil layer. In contrast, several studies recently investigated liquefaction behavior while considering biaxial effects using experimental and numerical approaches [17–21]. They studied the relationship between detailed properties of the soil and the liquefaction pattern according to the occurrence of excess pore water pressure and found important results such as the effects of load non-proportionality and a direct function of the phase angle of the induced shear stresses on pore water pressure buildup.

The results of the studies described above are meaningful in helping us understand the reliquefaction behavior following the occurrence of aftershocks and recent advances on soil liquefaction. However, considering that the greatest damage in the event of liquefaction and reliquefaction is mainly due to settlement or collapse of superstructures, there is a limitation because these studies treated the behavior of the soil itself and not of the soil with structures such as an embankment. Several previous studies employed a numerical approach to investigate the seismic behavior for the liquefiable soil condition of offshore structures such as breakwaters and pipelines [22,23]. Other studies performed numerical or experimental studies on earthquake behavior of embankments and identified that the widespread damage to such piles and embankments occurred mainly due to the liquefaction of foundation soil, resulting in excessive settlements, lateral spreading, and

slope instability [24–31]. However, the key to the above studies was to investigate the seismic behavior of the embankments or other coastal structures, which was far from the reliquefaction behavior of embankment structures, which play an important role in securing the stability of major infrastructure. Therefore, it is necessary to investigate reliquefaction in scenarios involving structures such as embankments.

In this study, the reliquefaction behavior characteristics of coastal embankment structures, which have location characteristics with great concerns regarding liquefaction and reliquefaction occurrences and may cause great damage, were investigated. As demonstrated in this study based on a shaking table that could simulate liquefaction, the thicknesses of the liquefiable and non-liquefiable layers in the ground on which an embankment was installed to obtain shaking load data for each ground layer when reliquefaction occurred were used to confirm the natural frequency and time history graphs of the response data. Additionally, the excess pore water pressure ratios were calculated from the first to fifth liquefaction cycles using a pore water pressure transducer, and a comparative analysis was conducted on the correlation between the acceleration and excess pore water pressure ratio during reliquefaction.

## 2. Liquefaction and Reliquefaction Mechanisms

Liquefaction can be classified into flow liquefaction and cyclic mobility. Flow liquefaction can occur when the static shear stress exceeds the steady-state strength. This phenomenon occurs primarily on slopes and causes flow failure, which is the most dangerous form of liquefaction-related damage. Cyclic mobility occurs when the static shear stress is less than the shear strength of the liquefiable ground. It mainly occurs in coastal areas with gentle slopes and when shaking occurs in uncompacted sandy soil with a short sedimentation age, such as saturated sand [32].

A characteristic of soil reliquefaction behavior is that the liquefaction resistance can decrease rapidly although the soil density increases as induced by drainage after liquefaction. Oda et al. reported that when liquefaction occurs in the soil, the grain structure of the soil is rendered unstable due to shear deformation [13]. Accordingly, excess pore water pressure increases abruptly, thereby allowing for liquefaction to occur more readily even due to small earthquakes. Ha compared the number of loading cycles with the change in excess pore water pressure during reliquefaction and confirmed that fewer load repetitions were required to trigger reliquefaction when compared with the initial liquefaction [33].

## 3. Materials and Methods

### 3.1. Ground Properties and Embankment Specifications

Silica sand No. 7 was used for the ground composition of the shaking table test. Figure 1 presents the grain size distribution of silica sand No. 7, which is included in the liquefaction hazard range (grain size range: 0.01–1.0 mm) suggested by the Applied Technology Council [34]. Table 1 lists the physical properties of the silica sand. The specific gravity (Gs) of the silica sand No. 7 used in this test was 2.65, the maximum dry unit weight was 18.17 kN/m$^3$, and the minimum unit dry weight was 13.47 kN/m$^3$. The dimensions of the model embankment were determined virtually based on the coastal embankment located in the S-Project in South Korea. Three types of similitude law were suggested by Iai [35]. Given that liquefaction closely resembles the strain-softening behavior, the third form of the similitude law was applied, and the ratio of the similitude was 40. Table 2 lists the properties of the embankment models.

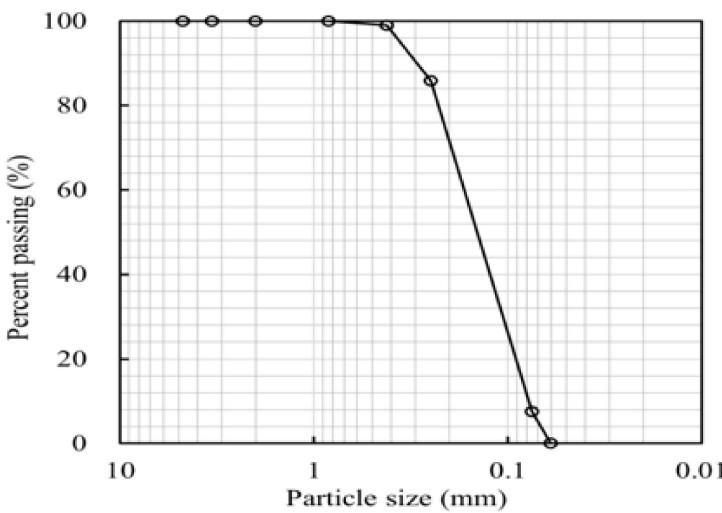

**Figure 1.** Particle size distribution curve for model soil.

**Table 1.** Soil properties of the model soil.

| Parameter | Value |
|---|---|
| Specific gravity, $G_s$ | 2.65 |
| Maximum void ratio, $e_{max}$ | 0.93 |
| Minimum void ratio, $e_{min}$ | 0.43 |
| Relative density, $D_r(\%)$ | 50 |
| Residual friction angle, $\varnothing(°)$ | 30.5 |
| Mean particle size, $D_{50}(mm)$ | 0.11 |
| Uniformity coefficient, $C_u$ | 2.89 |
| Coefficient of curvature, $C_c$ | 1.07 |
| Permeability, k (m/s) | $2.51 \times 10^{-4}$ |

**Table 2.** Mechanical properties of the embankment model.

| Parameter | Prototype | Model | Similitude Relationship |
|---|---|---|---|
| Top (mm) | 14,000 | 350 | $\lambda$ |
| Bottom (mm) | 28,400 | 710 | $\lambda$ |
| Height (mm) | 4000 | 100 | $\lambda$ |
| Length (mm) | 20,000 | 500 | $\lambda$ |
| Volume (cm$^3$) | $1.696 \times 10^9$ | 26,500 | $\lambda^3$ |
| Density (kg/m$^3$) | 2000 | 2000 | 1 |
| Load (kg) | 3,392,000 | 53 | $\lambda^3$ |
| Stress (kgf/m$^2$) | 5971.83 | 149.30 | $\lambda$ |

*3.2. Seismic Waves*

Figure 2 presents the input base motion profile, which was measured on a shaking table. In the shaking table test, a sinusoidal wave of 5 Hz was determined as the input motion, which corresponded to a sinusoidal wave of 0.8 Hz at the prototype scale while applying Iai's type 3 similitude relationship. Each sine wave excitation had a duration of 8 s; namely, 1.5 s for the increasing section, 5.0 s for the constant section, and 1.5 s for the decreasing section with an input acceleration of 0.2 g, which is the return period of a 2400-year earthquake in the Korean seismic design code. A total of five vibrations were excited to analyze the reliquefaction behavior due to successive earthquakes. The excitation interval was set to 1800 s so that the excess pore water pressure could be sufficiently dissipated. As a result of observing the dissipation of the excess pore water pressure through the measured pore water pressure with the pore water pressure transducer, it was confirmed that the excess pore water pressure converged to zero at around 600 s.

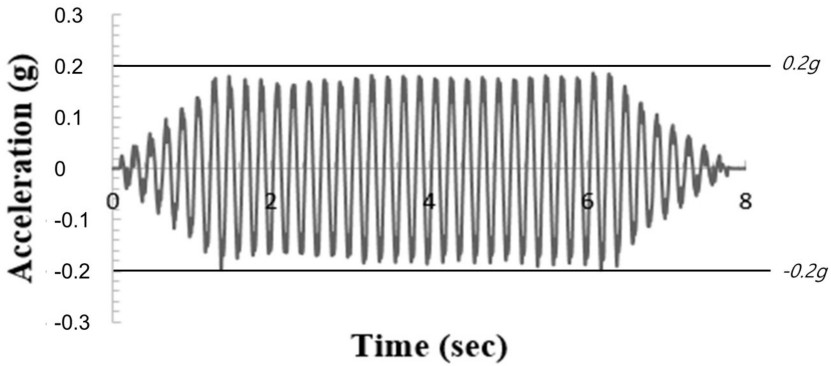

**Figure 2.** Base motion at 0.2 g and 5 Hz.

*3.3. Experimental Method*

To analyze the behavior of the coastal embankment with respect to the thickness of the non-liquefiable layer, the soil compositions were divided into two cases as follows. In Case 1, the soil layer comprised only a 50 cm liquefiable layer, and in Case 2, there were two layers; i.e., a 32.5 cm lower liquefiable layer and a 17.5 cm upper non-liquefiable layer. For each experiment, a reliquefaction test was conducted with five sine wave excitations using a shaking table. Gravel with a size ranging from 1 to 2 cm and a unit weight of 2.0 kN/m$^3$ was used for the embankment, which had a height of 10 cm and a fixed slope ratio of 1:1.8. The model simulated an actual embankment at a scale of 1:40. Given that an embankment is a fill structure on a road surface, an overload of 53 kg was applied while considering the actual weight of the section. To form the ground of the liquefiable layer, a sieve was installed on the soil box. Next, the soil particles were separated as evenly as possible; they were slowly dropped into the water to create a composition similar to the formation principle of the sedimentary layer, and the silica sand was saturated in water for 72 h. The model ground was formed at a relative density of 50%. The relative density was measured during a preliminary test. A sample could be placed every 10 cm from the bottom of the soil box when preparing the model ground. The ground composition was stopped; the sample was removed; and the weight, volume, and water content were measured to calculate the unit weight and relative density when the ground composition was completed. Afterwards, the model ground was formed in the same way, and the target relative density was constructed by making the weight of soil used in ground composition the same as that in the preliminary test. The non-liquefiable layer formed a total liquefiable ground, and then a hose was installed and dewatered after excavating so that the location did not interfere with the installation of the embankment and instrumentation to create the non-liquefiable layer. Figure 3 illustrates the procedure employed for the experimental setup. The test was conducted using a soil chamber with a length, width, and height of 200 cm, 50 cm, and 70 cm, respectively. To reduce the boundary effect of waves due to the stiffness of the soil chamber during shaking, a 5 cm thick sponge was installed on both walls of the soil chamber. Figures 4 and 5 present the cross sections in the experiment, including the measuring instrument. To analyze the ground behavior during reliquefaction and the occurrence of liquefaction with respect to depth, piezometers were installed at the end of the embankment, at the center of the embankment, and in the free field at depths of approximately 10 cm, 20 cm, 30 cm, 40 cm, and 45 cm from the ground surface. The piezometers were fixed using aluminum rods to maintain a constant height in the liquefiable soil. Accelerometers were installed at depths of 0, 10, 20, 30, 40, and 50 cm from the ground surface and fixed to a square acrylic plate to ensure soil-like behavior when liquefaction occurred. Linear variable differential transformers (LVDTs) were installed at the center of the embankment and free field to measure the amount of settlement in the embankment and free field. The test programs are summarized in Table 3.

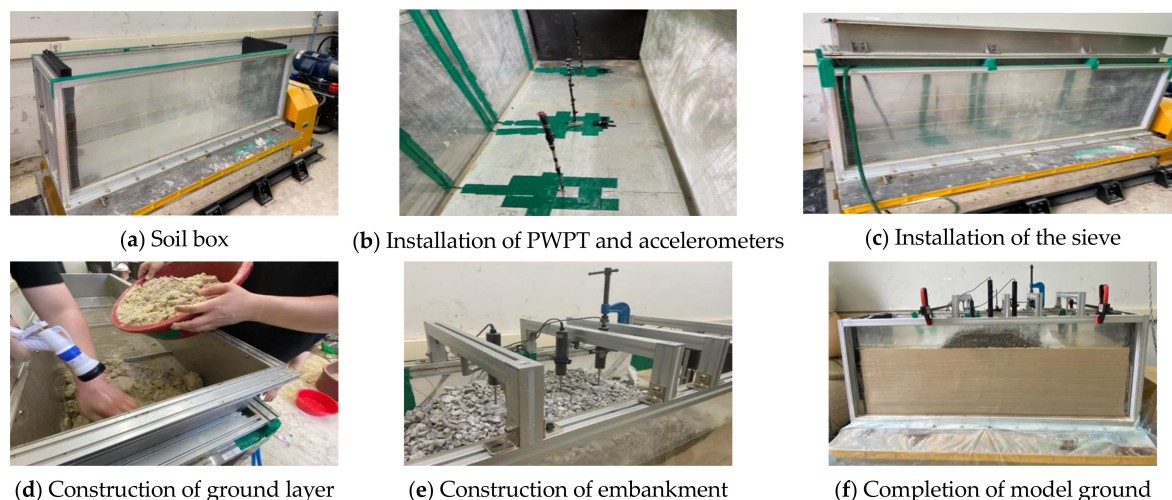

(**a**) Soil box     (**b**) Installation of PWPT and accelerometers     (**c**) Installation of the sieve

(**d**) Construction of ground layer     (**e**) Construction of embankment     (**f**) Completion of model ground

**Figure 3.** Test setup.

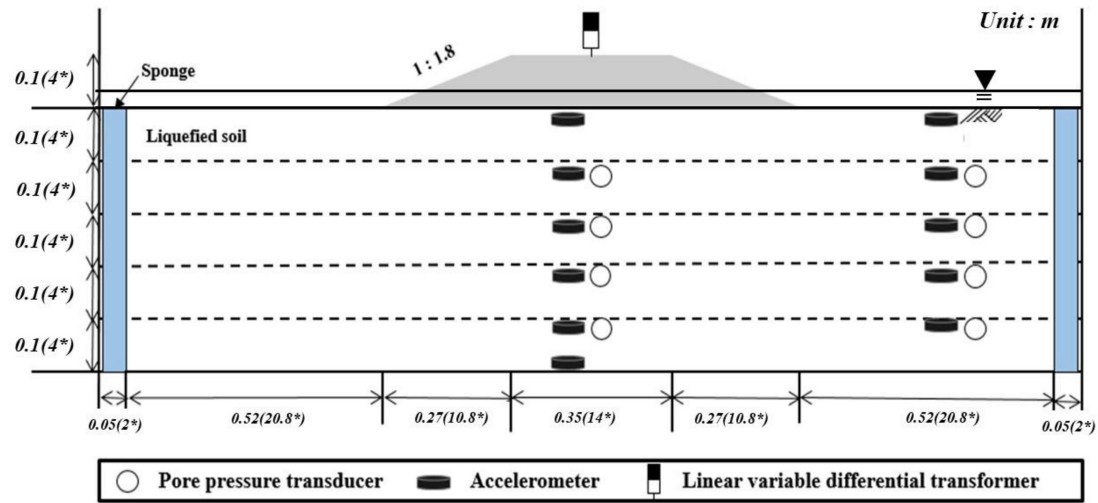

**Figure 4.** Schematic drawing of test section of liquefiable ground for model scale of Case 1 (* prototype scale).

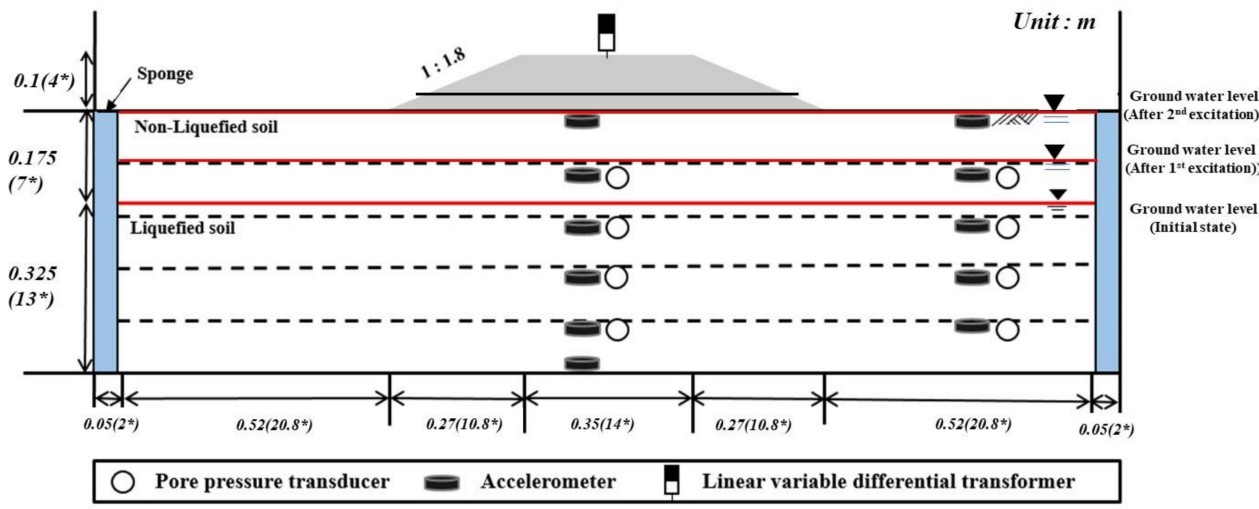

**Figure 5.** Schematic drawing of test section of non-liquefiable and liquefiable ground for model scale of Case 2 (* prototype scale).

**Table 3.** Test program.

| Case | Composition of Facilities | Thickness of the Lower Liquefiable Layer (m) | Thickness of the Upper Non-Liquefiable Layer (m) | Note |
|---|---|---|---|---|
| Case 1 | Embankment (h = 0.1 m, slope: 1:1.8 fixed) | 0.500 (20) * | 0 | Performance with 1–5 vibrations to confirm reliquefaction behavior |
| Case 2 | | 0.325 (12.5) * | 0.175 (7.5) * | |

* Prototype properties.

## 4. Results and Discussion

Based on the shaking table tests, the accelerometers, piezometers, and LVDTs installed in each layer were used to calculate the acceleration–time history, excess pore water pressure ratio, embankment settlement amount, and relative density. The results presented in the subsequent sections were based on a scaled embankment model (1:40) experiment scaled to the prototype size using the third form of the Iai similitude.

### 4.1. Liquefiable Ground Case (Case 1)

All test results were described at the prototype scale by applying a similitude ratio. To obtain data on the shaking load for each ground layer and check the state of liquefaction, 11 accelerometers and 8 pore water pressure transducers were installed for the experiment as shown in Figure 4. Figure 6 presents the acceleration–time history measured during the first excitation using accelerometers installed on the ground surface and at a depth of 16 m in the free field. The acceleration–time history also indicates the occurrence of liquefaction. Figure 6a reveals that the acceleration decreased rapidly due to the occurrence of liquefaction at the ground surface below the embankment. Given that the ground behaved similarly to a liquid when liquefaction occurred, the ground reaction did not occur and the amplitude decreased [36]. In contrast, liquefaction did not occur at a depth of 16 m; therefore, the acceleration value did not decrease significantly. It was also confirmed by the excess pore water pressure ratio and acceleration that during the first excitation, liquefaction occurred from the ground surface to a depth of 8 m and not more than a depth of 12 m from the ground surface (Figure 7).

The excess pore water pressure ratio was calculated by dividing the excess pore water pressure generated over time by the effective stress. The effective stress below the embankment was calculated by applying an additional surcharge of the embankment body. Based on previous research, the occurrence of liquefaction is determined when the excess pore water pressure exceeds 1.0 [37–39]. The liquefied ground was determined when the excess pore water pressure increased to a value larger than 1.0. Figures 7–9 present a representative graph of the excess pore water pressure ratio with a depth below the center of the embankment and the free field due to successive excitations; in addition, the maximum values of the excess pore water pressures in all cases are described in Tables 4 and 5. The porewater pressure transducer installed at a 8 m depth in the free field did not work because of technical issues; therefore, measurements could not be performed. Below the center of the embankment (as shown in Figure 7), the excess pore water pressure ratios at depths of 4 m and 8 m exceeded unity, indicating that liquefaction occurred. The excess pore water pressure ratios at depths of 12 m and 16 m were less than 1 as listed in Figure 7a and Table 5. These findings indicated that liquefaction did not occur. However, according to the piezometers located in the free field (Figure 7b), liquefaction occurred at all depths during the first excitation. This indicated that due to the overload pressure caused by embankment subsidence, the ground-confining pressure increased more than that in the free field, leading to different trends from those obtained at a depth of more than 12 m.

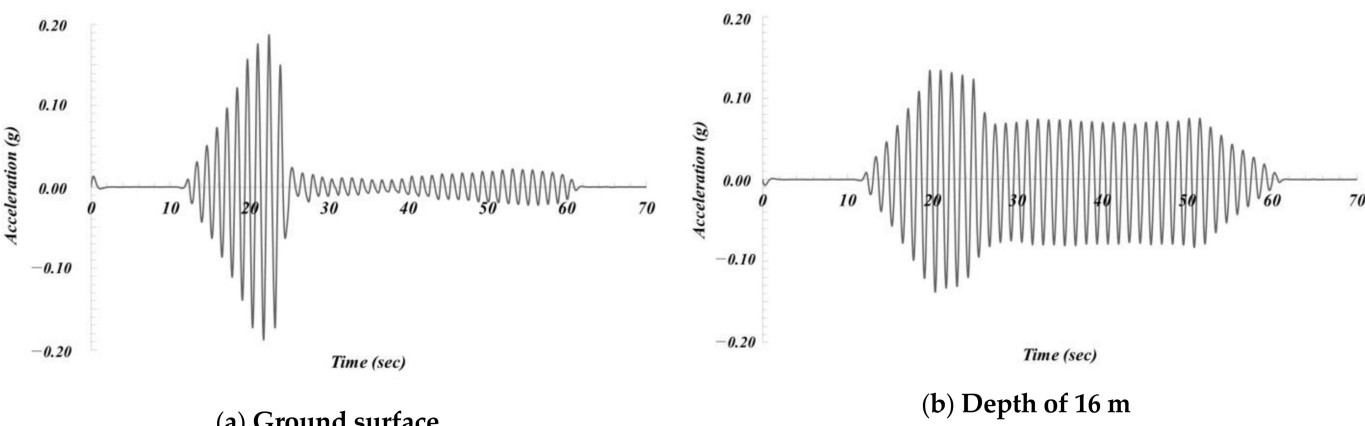

(a) **Ground surface**  (b) **Depth of 16 m**

**Figure 6.** Measured acceleration–time history for first shaking event in Case 1.

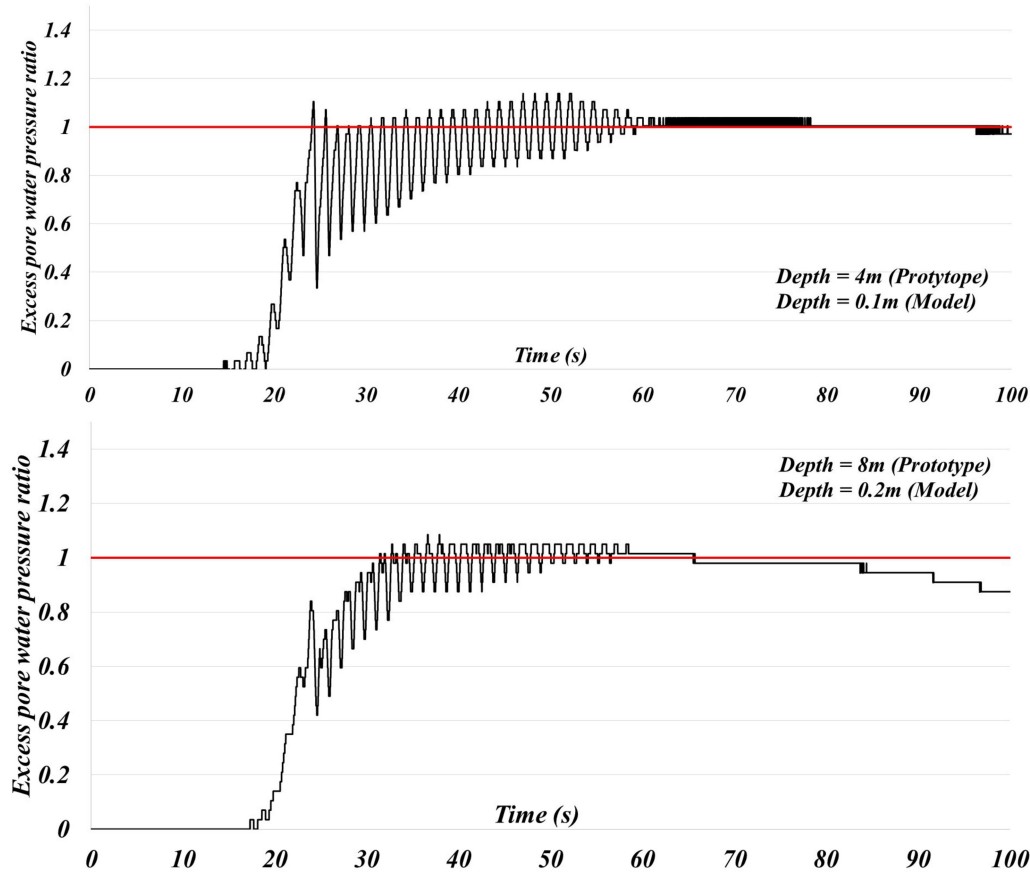

**Figure 7.** *Cont.*

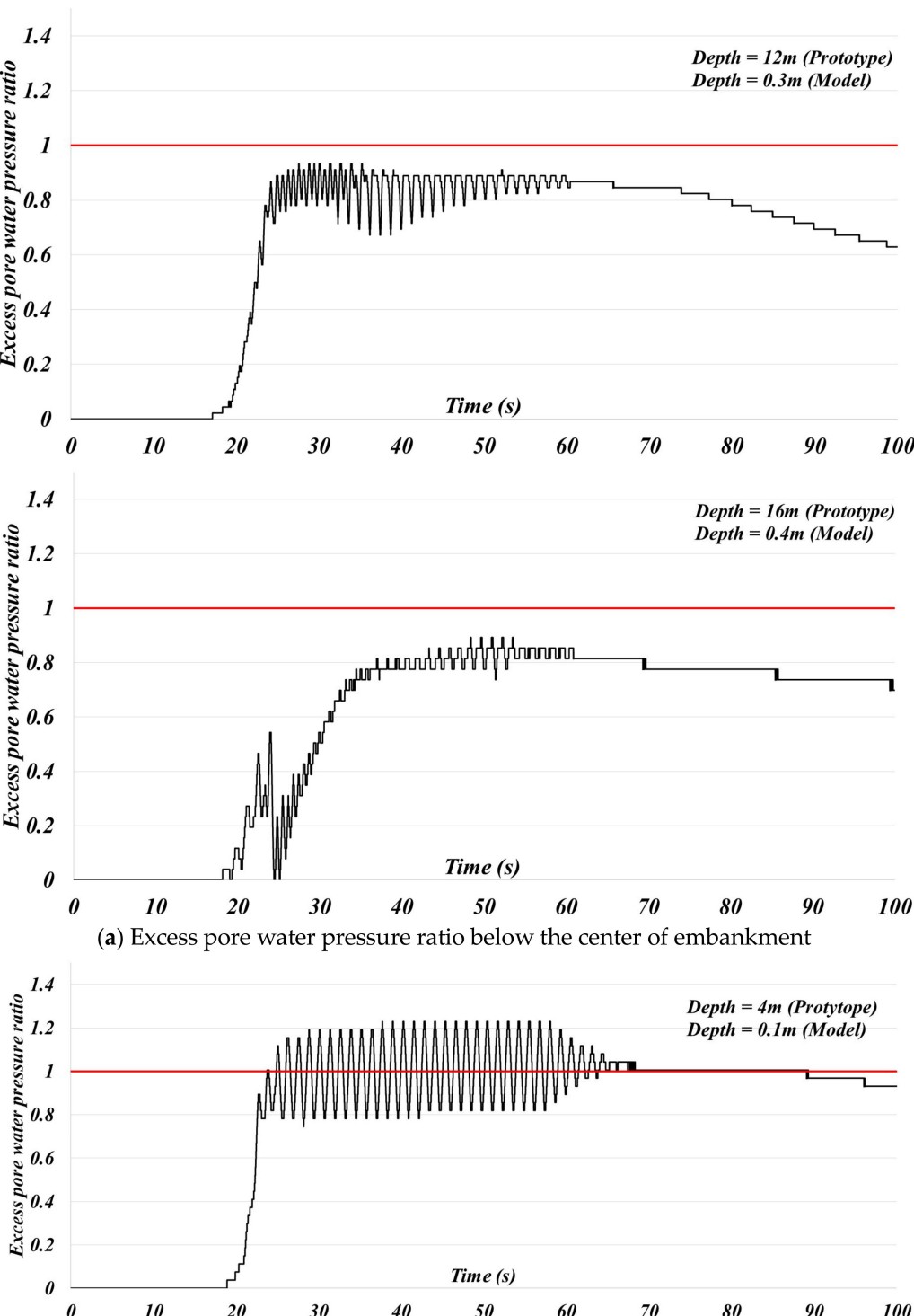

(**a**) Excess pore water pressure ratio below the center of embankment

**Figure 7.** *Cont.*

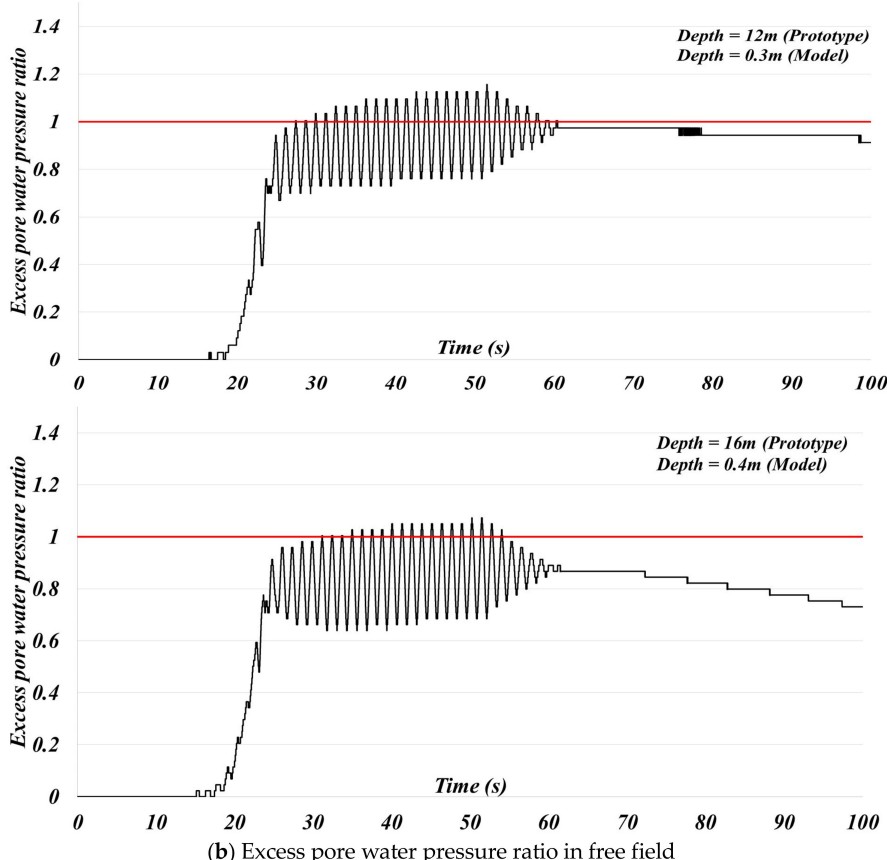

(**b**) Excess pore water pressure ratio in free field

**Figure 7.** Excess pore water pressure ratio for first shaking event (Case 1).

**Table 4.** Excess pore water pressure ratio at the center of the embankment (Case 1).

| Event Number | Excess Pore Water Pressure Ratio (Depth: 4 m) | Excess Pore Water Pressure Ratio (Depth: 8 m) | Excess Pore Water Pressure Ratio (Depth: 12 m) | Excess Pore Water Pressure Ratio (Depth: 16 m) |
|---|---|---|---|---|
| First | 1.14 | 1.09 | 0.93 | 0.85 |
| Second | 1.01 | 0.85 | 0.87 | 0.80 |
| Third | 0.84 | 0.73 | 0.71 | 0.70 |
| Fourth | 0.70 | 0.65 | 0.62 | 0.62 |
| Fifth | 0.54 | 0.60 | 0.57 | 0.57 |

**Table 5.** Excess pore water pressure ratio in the free field (Case 1).

| Event Number | Excess Pore Water Pressure Ratio (Depth: 4 m) | Excess Pore Water Pressure Ratio (Depth: 8 m) | Excess Pore Water Pressure Ratio (Depth: 12 m) | Excess Pore Water Pressure Ratio (Depth: 16 m) |
|---|---|---|---|---|
| First | 1.22 | | 1.15 | 1.07 |
| Second | 1.15 | | 1.05 | 0.92 |
| Third | 1.02 | N/A | 0.87 | 0.84 |
| Fourth | 0.92 | | 0.85 | 0.73 |
| Fifth | 0.82 | | 0.72 | 0.68 |

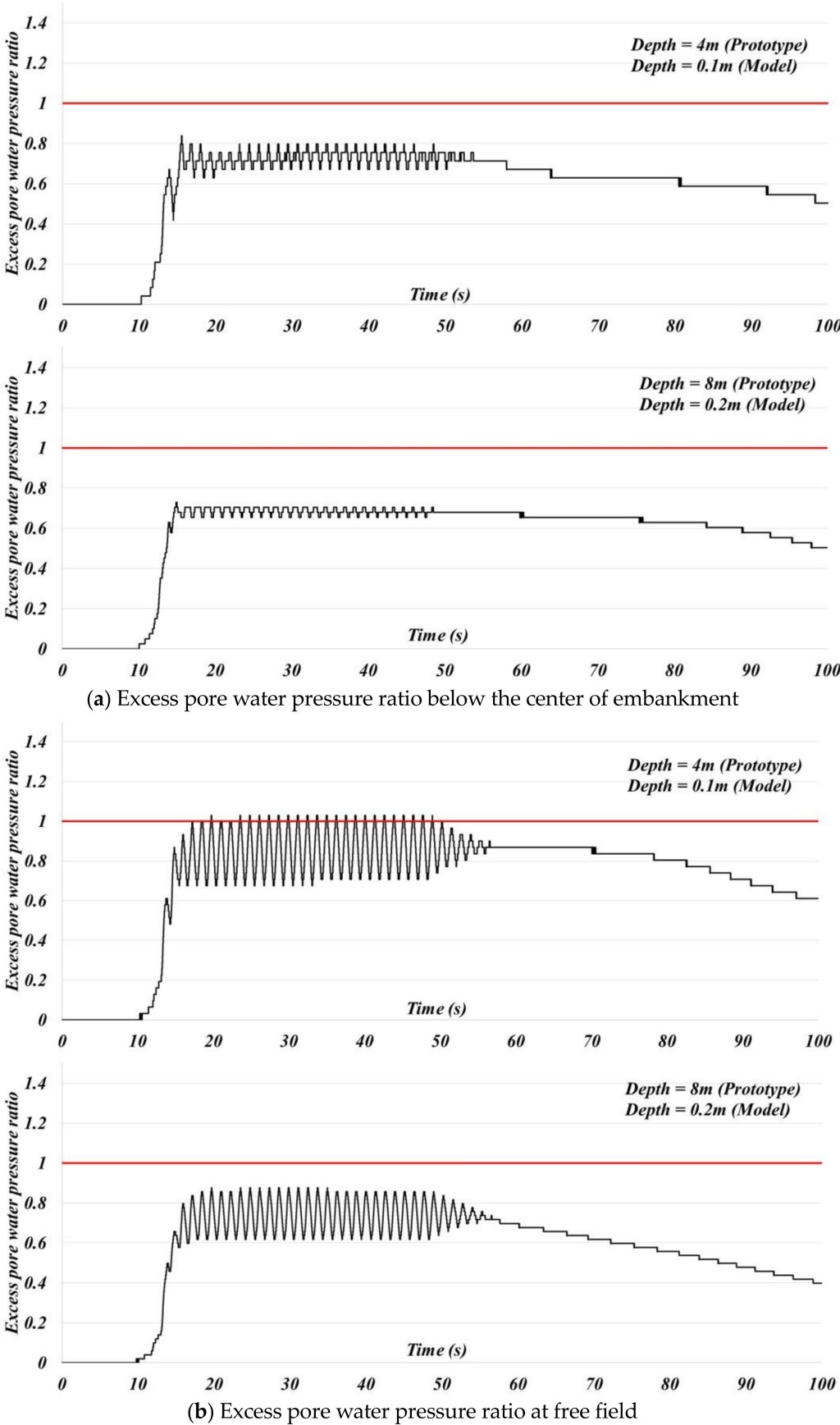

(**a**) Excess pore water pressure ratio below the center of embankment

(**b**) Excess pore water pressure ratio at free field

**Figure 8.** Excess pore water pressure ratio for the third shaking event (Case 1).

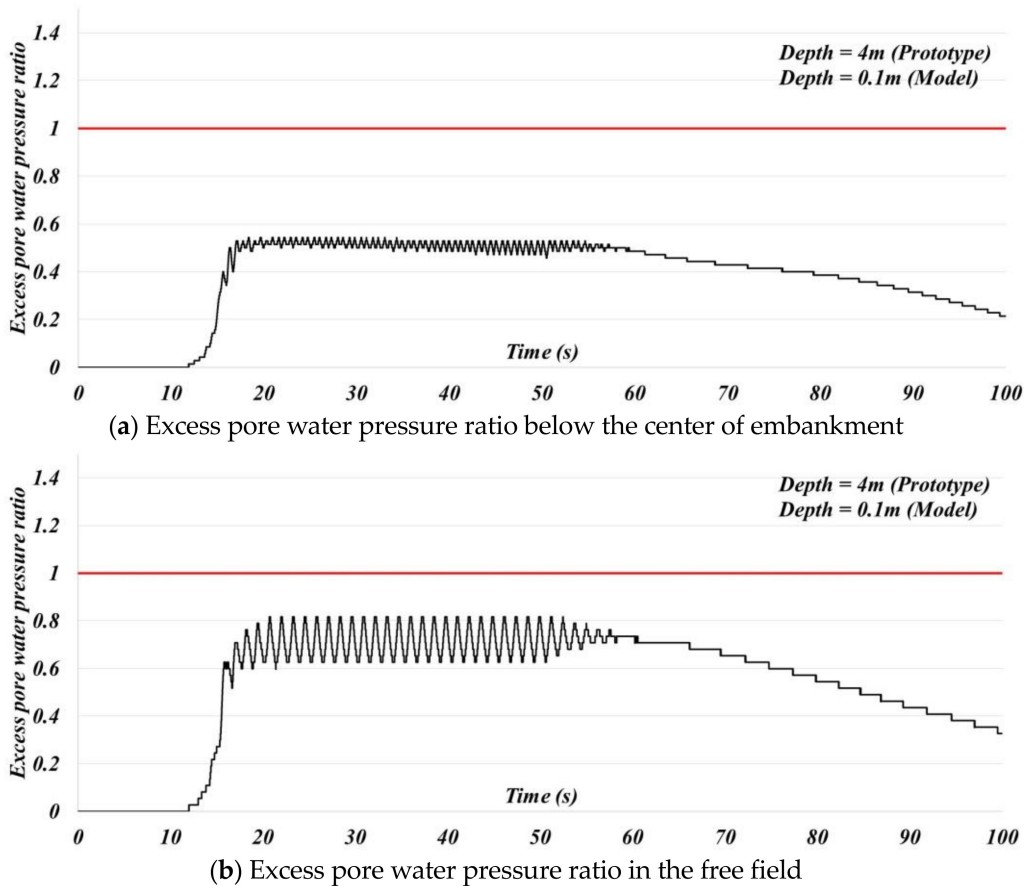

(**a**) Excess pore water pressure ratio below the center of embankment

(**b**) Excess pore water pressure ratio in the free field

**Figure 9.** Excess pore water pressure ratio for the fifth shaking event (Case 1).

Figure 8 shows the pore water pressure ratio for the third shaking event. As shown in Figure 8a, liquefaction did not occur at the center of the embankment during the third shaking excitation due to densification of the ground and confining pressure of the embankment structure. As shown in Figure 8b, liquefaction only occurred at a depth of 4 m in the free field.

During the fifth excitation, the excess pore water pressure ratios below the center of the embankment and free field were less than unity at all depths, indicating that liquefaction did not occur (Figure 9, Tables 4 and 5) because of densification of the ground during successive earthquakes. These findings suggested that liquefaction did not occur because the relative density increased due to ground subsidence caused by the overload pressure from the embankment and repeated shaking. In addition, as shown in Table 5, the maximum excess pore water pressure at a depth of 8 m was lower than that at a depth of 4 m below the center of embankment by the fifth earthquake. This meant that the increase in the confining pressure and the densification of the ground occurred more in the shallow ground due to the settlement of the embankment structure due to repeated earthquakes.

As shown in Figure 10, settlement (as measured using the LVDTs) rapidly occurred during the first excitation and substantially decreased from the first to fifth excitations. The settlement increased while the excitation continued, and after excitation ended, little additional settlement occurred even though the excess pore water pressure still remained. This result was consistent with that of a previous study [40]. Table 6 presents the data for the settlement amount and relative density according to the number of excitations. The relative density was calculated using the value obtained by dividing the total volume and mass of the ground while considering the settlement. Although the relative density should be presented with respect to each ground depth and excitation step, due to experimental limitations, this paper presents the relative density for the entire ground. These findings suggested that the relative density increased to 67.7% during the fifth excitation; that as the

number of excitations increased, the settlement amount decreased; and that liquefaction did not occur in the lower ground in the free field. In addition, when the relative density reached 60%, the excess pore water pressure ratio did not exceed unity except at a depth of 4 m; therefore, further liquefaction did not occur below the embankment under an excitation level of 0.2 g.

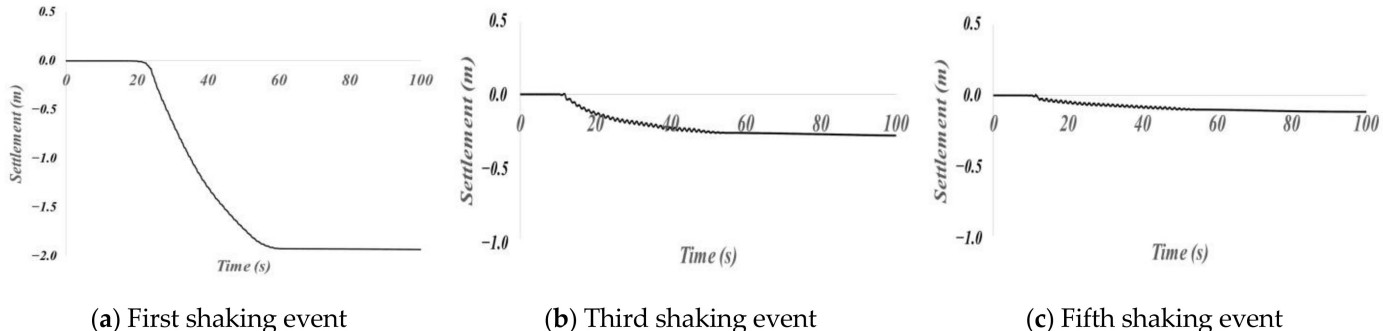

| (**a**) First shaking event | (**b**) Third shaking event | (**c**) Fifth shaking event |
|---|---|---|

**Figure 10.** LVDT at the center of the embankment (Case 1).

**Table 6.** Relative density and ground settlement at the center of the embankment (Case 1).

| Event Number | Shake 1 | Shake 2 | Shake 3 | Shake 4 | Shake 5 |
|---|---|---|---|---|---|
| Settlement (prototype) | 1.96 | 0.78 | 0.29 | 0.15 | 0.12 |
| (Accumulated settlement), m | (1.96) | (2.74) | (3.03) | (3.18) | (3.3) |
| Settlement (model) | 0.049 | 0.019 | 0.007 | 0.004 | 0.003 |
| (Accumulated settlement), m | (0.049) | (0.068) | (0.075) | (0.079) | (0.082) |
| Relative density, % | 59.4 | 63.1 | 64.5 | 65.3 | 65.8 |

Figure 11 presents the correlation between the excess pore water pressure ratio and the relative density at different depths below the center of the embankment for comparison. The relative density gradually increased as the number of excitations increased, and liquefaction occurred in the first excitation at depths of 4 m and 8 m as the excess pore water pressure ratio exceeded 1. However, liquefaction did not occur at depths of 12 m and 16 m when the excess pore water pressure ratio was less than 1. In addition to the non-occurrence of liquefaction, the increasing trend of relative density declined. When the relative density exceeded approximately 65–66%, the excess pore water pressure ratio decreased to values less than 1 at all depths. These findings suggested that as the number of excitations increased and the relative density reached a certain value, the ground density increased; thus, the excess pore water pressure ratio did not exceed 1, and liquefaction did not occur.

*4.2. Liquefiable and Non-Liquefiable Ground Case (Case 2)*

When there was an upper non-liquefiable layer in the ground as shown in Figure 12, liquefaction did not occur in any of the ground layers during the first excitations. Figure 13 presents a representative graph of the excess pore water pressure ratio with a depth below the center of the embankment caused by successive excitation; in addition, the maximum values of the excess pore water pressure in all cases are described in Tables 7 and 8. In Case 2, the excess pore water pressure ratio was calculated by considering the change of effective stress due to the groundwater level rise.

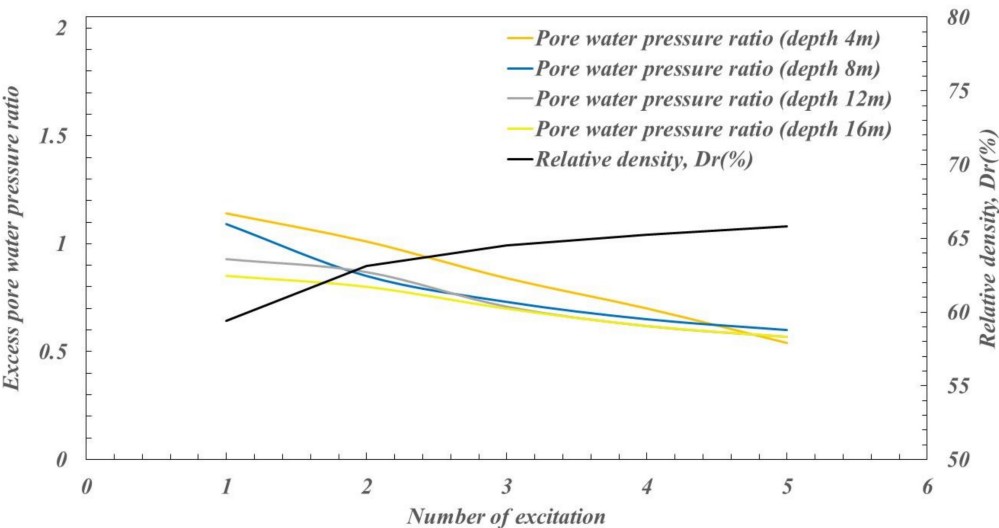

**Figure 11.** Excess pore water pressure ratio and relative density (center of the embankment, Case 1).

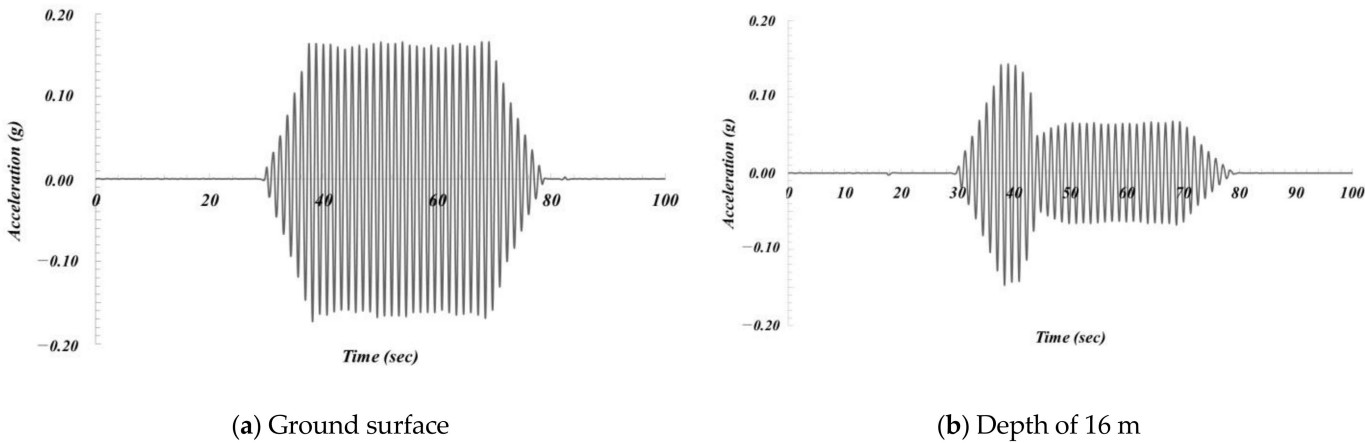

(**a**) Ground surface                        (**b**) Depth of 16 m

**Figure 12.** Measured acceleration time history for first shaking event in Case 2.

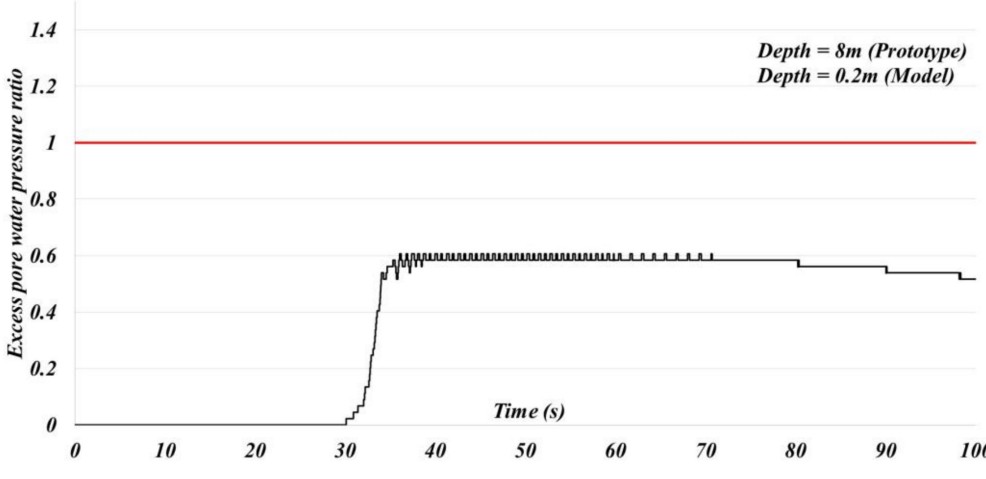

**Figure 13.** *Cont.*

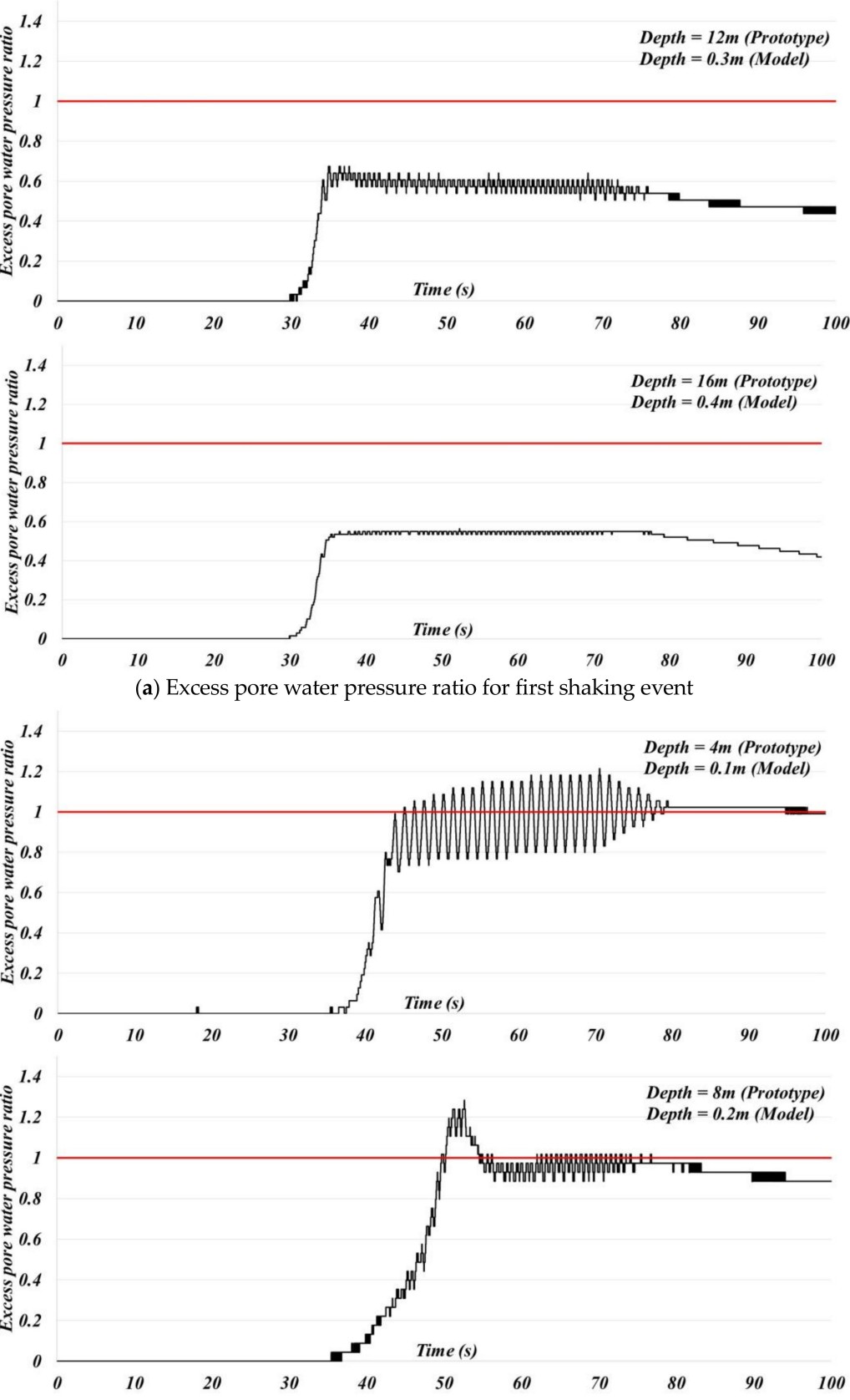

(**a**) Excess pore water pressure ratio for first shaking event

**Figure 13.** *Cont.*

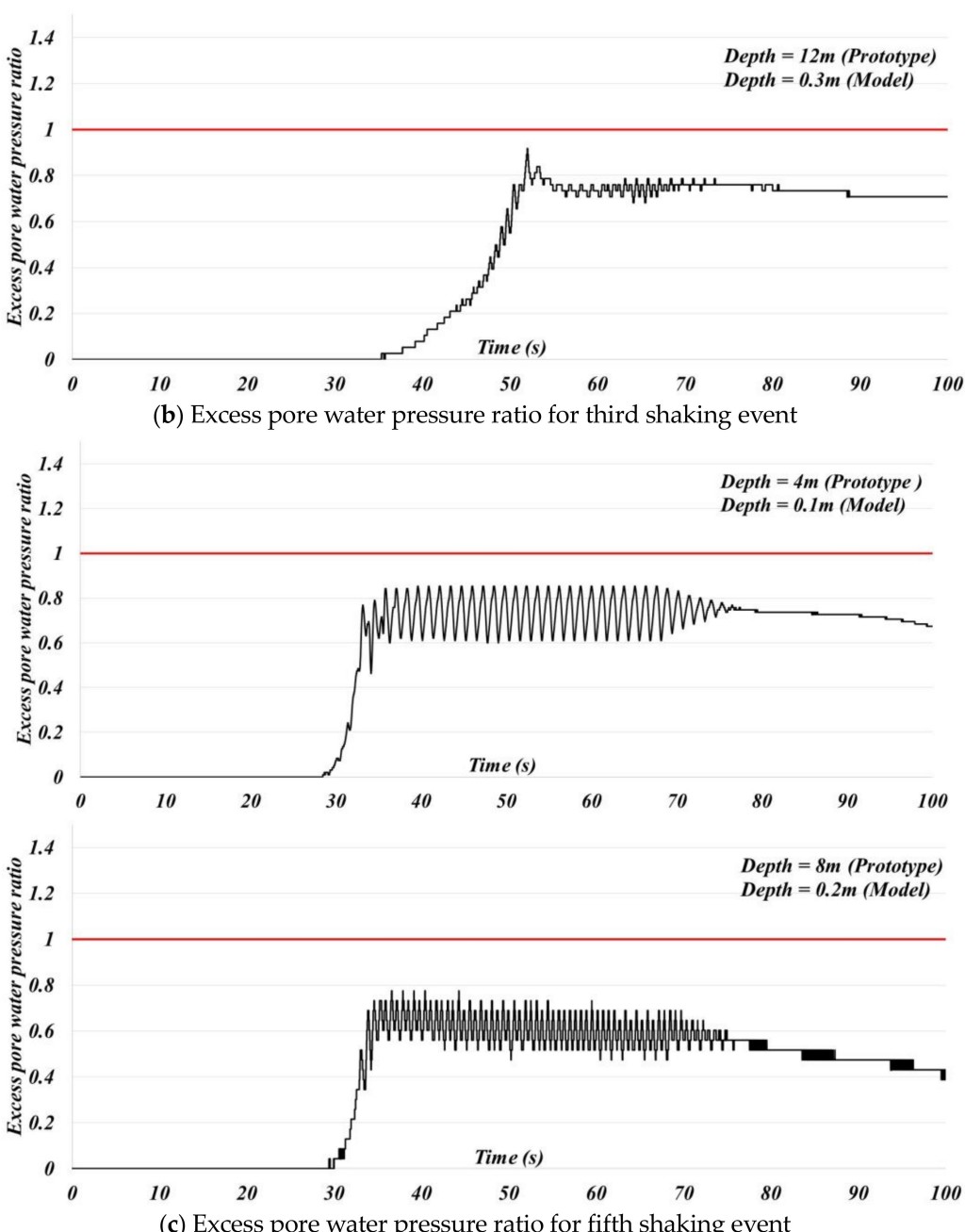

(**b**) Excess pore water pressure ratio for third shaking event

(**c**) Excess pore water pressure ratio for fifth shaking event

**Figure 13.** Excess pore water pressure ratio for successive shaking events below the center of the embankment (Case 2).

**Table 7.** Excess pore water pressure ratio at the center of the embankment (Case 2).

| Event Number | Excess Pore Water Pressure Ratio (Depth: 4 m) | Excess Pore Water Pressure Ratio (Depth: 8 m) | Excess Pore Water Pressure Ratio (Depth: 12 m) | Excess Pore Water Pressure Ratio (Depth: 16 m) |
|---|---|---|---|---|
| First | 0.00 | 0.60 | 0.67 | 0.56 |
| Second | 0.42 | 0.70 | 0.73 | 0.63 |
| Third | 1.21 | 1.28 | 0.92 | 0.81 |
| Fourth | 1.01 | 0.92 | 0.83 | 0.76 |
| Fifth | 0.85 | 0.78 | 0.76 | 0.68 |

**Table 8.** Excess pore water pressure ratio in the free field (Case 2).

| Event Number | Excess Pore Water Pressure Ratio (Depth: 4 m) | Excess Pore Water Pressure Ratio (Depth: 8 m) | Excess Pore Water Pressure Ratio (Depth: 12 m) | Excess Pore Water Pressure Ratio (Depth: 16 m) |
|---|---|---|---|---|
| First | 0.00 | | 0.83 | 0.63 |
| Second | 0.53 | | 0.91 | 0.78 |
| Third | 1.18 | N/A | 1.02 | 0.87 |
| Fourth | 1.07 | | 0.93 | 0.82 |
| Fifth | 0.91 | | 0.81 | 0.75 |

As shown in figure, liquefaction did not occur in the first excitation, whereas it occurred in the third excitation. The groundwater level rose to 4 m below the ground surface in the first earthquake and up to the ground surface in the second earthquake (Figure 5). These findings suggested that as the seismic load was applied, the groundwater level in the liquefiable layer increased to the height of the non-liquefiable layer, the entire ground was submerged in groundwater, and liquefaction occurred in the third excitation onward. To confirm the effects of reliquefaction, a total of five excitations were performed. According to Table 7, the excess pore water pressure ratio exceeded 1.0 above the layer of 8 m in the third and above the layer of 4 m in the fourth excitation, indicating that liquefaction occurred. In addition, even in the layer where liquefaction did not occur, excess pore water pressure values were observed in the fourth and fifth excitation that were larger than those after the first earthquake because the effect of the confining pressure was reduced due to an increase in the groundwater level. As shown in Table 8, when there was a non-liquefiable layer, liquefaction occurred above 12 m in the third excitations and above 4 m in the fourth excitation in the free field and not in the fifth excitations. The difference in the occurrence of liquefaction between the free field and below the center of the embankment mirrored the observations from Case 1. In particular, liquefaction did not occur in the third excitation at a depth of 12 m when there was an embankment; this was due to the overload confining pressure, which was in contrast to the ground case without an embankment.

Figure 14 presents a graph of the amount of embankment settlement based on the LVDT installed at the center of the embankment. Table 9 presents the data for the settlement amount and relative density according to the number of excitations. The change in relative density was not significant in the first and second excitations because liquefaction did not occur in the saturated ground due to the relatively high confining pressure caused by non-liquefiable ground. In addition, the groundwater level increased, and the entire ground was submerged in groundwater; it was judged that the upward pressure of groundwater level led to a relatively low settlement. The relative density remained similar; therefore, liquefaction occurred from the third excitation. Subsequently, the relative density increased, and the amount of settlement in each excitation decreased in subsequent excitations.

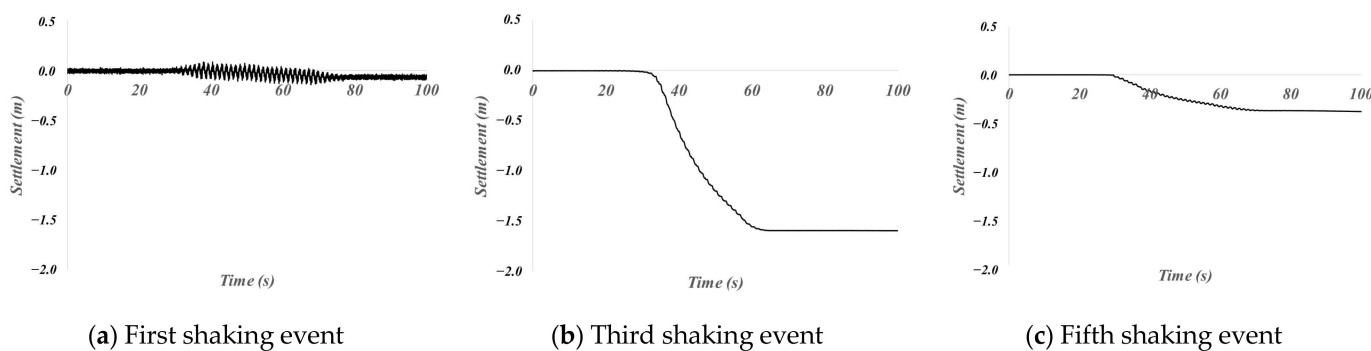

(**a**) First shaking event  (**b**) Third shaking event  (**c**) Fifth shaking event

**Figure 14.** LVDT at the center of the embankment (Case 1).

**Table 9.** Relative density and ground settlement at the center of the embankment (Case 2).

| Event Number | Shake 1 | Shake 2 | Shake 3 | Shake 4 | Shake 5 |
|---|---|---|---|---|---|
| Settlement (prototype) (Accumulated settlement), m | 0.13 (0.13) | 0.25 (0.38) | 1.59 (1.97) | 0.51 (2.48) | 0.40 (2.88) |
| Settlement (model) (Accumulated settlement), m | 0.003 (0.003) | 0.007 (0.010) | 0.040 (0.050) | 0.013 (0.063) | 0.010 (0.073) |
| Relative density, % | 50.6 | 51.9 | 59.5 | 61.9 | 63.9 |

Figure 15 presents a graph of the correlation between the excess pore water pressure ratio and the relative density at different depths below the center of the embankment when there was an upper non-liquefiable layer in the ground. Liquefaction did not occur until the third excitation when there was a non-liquefiable layer. Thus, no significant changes were observed in the relative density or excess pore water pressure ratio. However, liquefaction occurred during the third excitation, and the excess pore water pressure ratio exceeded 1.0, demonstrating an increasing trend similar to that of the relative density. Furthermore, after the relative density reached 60% or greater, the excess pore water pressure ratio did not exceed unity, indicating that further liquefaction did not occur under an excitation level of 0.2 g. It was determined that this phenomenon could occur when the thickness of the non-liquefaction layer was relatively low and the thickness of the liquefiable layer was more than twice the thickness of the non-liquefaction layer.

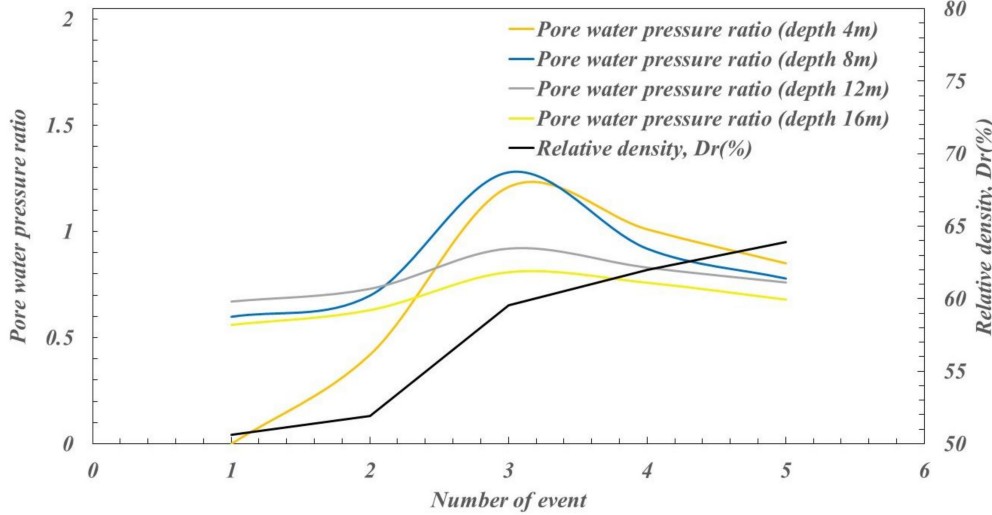

**Figure 15.** Excess pore water pressure ratio and relative density (center of the embankment, Case 2).

Based on a series of results, it was confirmed that the risk of liquefaction due to aftershocks is greater than that of the main earthquake in the case of a coastal embankment where the groundwater level is low (especially when the relative density of the ground is lower than 60%). In the case of general seismic design criteria, liquefaction risk assessment is not performed for soil layers higher than the groundwater level; however, if repeated aftershocks occur, unexpected liquefaction damage may occur due to an increment in the groundwater level. Therefore, to mitigate the earthquake risk of liquefaction for coastal embankments, it is necessary to evaluate liquefaction by aftershocks even when the groundwater level of the ground layer under an embankment is low. However, since this study summarized the results of experiments performed for limited experimental cases, additional experimental and numerical analysis studies on various liquefaction layer thicknesses are needed.

## 5. Conclusions

In this study, a series of shaking table tests were conducted while considering the thicknesses of liquefiable and non-liquefiable layers in saturated sand upon which an embankment was installed. Accelerometers, piezometers, and LVDTs were used to analyze the occurrence of liquefaction and ground behavior with respect to the depth during reliquefaction. The findings of this study can be summarized as follows:

(1) In Case 1, the liquefaction occurred above 12 m in the first and the second excitations in the free field and only occurred at the depth of 4 m in the third excitation. At the center of the embankment, the excess pore water pressure ratio exceeded unity above 8 m in the first excitation and only reached unity at the depth of 4 m in the second excitation. In this regard, the difference in the confining pressure caused by the overload pressure from the embankment most probably influenced the occurrence of liquefaction.

(2) When the upper ground layer consisted of a non-liquefiable layer (Case 2), liquefaction did not occur in the first excitation and occurred in the third excitation. These results indicated that as the shaking load was applied, the water level in the liquefiable layer increased to the height of the non-liquefiable layer, and liquefaction occurred. This suggested that when there is a liquefiable layer under a non-liquefiable layer, liquefaction may occur due to aftershocks. In the case of general seismic design criteria, liquefaction risk assessment is not performed for soil layers higher than the groundwater level; however, if repeated aftershocks occur, unexpected liquefaction damage may occur due to an increment in the groundwater level.

(3) In Case 1, the excess pore water pressure ratio decreased below unity after a relative density of 65% at a depth of 4 m. Additionally, in ground with a non-liquefiable layer, the excess pore water pressure ratio decreased after a relative density of approximately 63%. In both cases, liquefaction did not occur when the relative density was approximately 65% or higher, which can serve as a basis for gauging the likelihood of liquefaction when the relative density reaches a certain value.

(4) In this study, it was confirmed that even if liquefaction does not occur at the main earthquake, liquefaction occurs due to aftershocks caused by a rise in the groundwater level. In a general seismic design criterion, liquefaction assessment is performed only for soil layers below the groundwater level; however, if aftershocks occur, unexpected liquefaction damage may occur to coastal embankments. Therefore, to mitigate the earthquake risk of liquefaction for coastal embankments, it is necessary to evaluate liquefaction due to aftershocks even when the groundwater level of the ground layer under an embankment is low.

For the results of this study to be applied quantitatively, additional research on groundwater level rise due to earthquakes and the evaluation of liquefaction of subsequent aftershocks should be conducted via dynamic centrifuge tests and numerical analysis. In addition, further studies on various soil types also should be conducted in order to derive effective results that can be applied to various field conditions.

**Author Contributions:** M.Y. organized the paperwork, produced the analysis plan, and performed the shaking table test and data analysis; S.Y.K. supported the test result analysis and paperwork organization. Both authors contributed to the writing of the paper. All authors have read and agreed to the published version of the manuscript.

**Funding:** This research was supported by a grant from the Korea Institute of Energy Technology Evaluation and Planning (KETEP) funded by the Korean government (MOTIE) (20203020020040, Development of environmental monitoring and information disclosure system for onshore wind farms for data-based environmental evaluation and enhancement of acceptance, 2023-011(R)), which was conducted by the Korea Environment Institute (KEI); and a grant (RS-2023-00238458, Development and Verification of Integrated Management System for High-Risk Disaster Response in Deep Railway Facilities) from the Development of a Disaster Response Complex Training Center for Deep Tunnels (GTX, etc.) Program funded by the Ministry of Land, Infrastructure, and Transport of the Korean government. We greatly appreciate the support.

**Institutional Review Board Statement:** Not applicable.

**Informed Consent Statement:** Not applicable.

**Data Availability Statement:** Not applicable.

**Conflicts of Interest:** The authors declare no conflict of interest.

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
