# Peer review of "Evaluation of Reliquefaction Behavior of Coastal Embankment Due to Successive Earthquakes Based on Shaking Table Tests"

_jmse, doi:10.3390/jmse11051002_

Round 1

Reviewer 1 Report

In this paper, authors conducted two sets of continuous vibration tests to examine the reliquefaction of sand under earthquake conditions. The liquefaction conditions of the embankment with those of the surrounding free field were compared. A scenario where a non-liquefied layer covers the upper part of the liquefied layer were analyzed. Based on the findings, conclusions about the excess pore water pressure and relative density were drew. In light of these results, I would like to propose some suggestions:

1. The introduction section lacks a comprehensive overview of the existing research. It is advisable to integrate the research background of the predecessors, explain how this article extends the research, and include additional references as appropriate.

2. The input ground motion depicted in Figure 2 look like to resemble the recorded data from the shaking table, it appears that the amplitude of the values does not reach 0.2g, particularly the values above the x-axis.

3. In L199 of 4.1, liquefaction did not transpire at the depth of 16m. Figure 6 displays the acceleration amplitude curves. Is it feasible to determine whether liquefaction has transpired based on acceleration amplitude? For instance, can we determine at what point the acceleration amplitude reduction leads to soil liquefaction, or the percentage by which it needs to decrease?

4. The analysis of the results employs the prototype size, as previously mentioned. However, L288 includes data such as 5cm and 20cm. It is necessary to verify if these measurements pertain to the prototype size or the model size.

5. The conclusion section is excessively verbose as it elaborates extensively on the test results already presented in Chapter 4. The conclusions should be succinct and rephrased.

6. The third and fourth points of the conclusion reveal that the author possesses a relatively lucid comprehension of the constraints in this paper. I propose that instead of listing these limitations in the conclusion, they be summarized after it.

L21: The first letter of the sentence is not capitalized

The overall quality of English is good.

Author Response

The authors would like to sincerely thank the reviewer for your valuable comments regarding the manuscript. Based on the comments, the authors have revised the manuscript. All revisions are provided in the text and detailed answers to each comment are provided as attachment.

Reviewer 2 Report

Please download the attachement!

Not applicable. I have no much time to access the English writting.

Author Response

(The authors gave the same response as above.)

Reviewer 3 Report

The paper is about re-liquefaction behavior of coastal embankment based on shaking table tests. The methodology and analysis results are well explained. Overall, the paper quality and write up is good, it just needs some modifications to be ready for publication. I am presenting my remarks and comments in details below:

1-    The paper writing has several awkward phrases, and the overall language quality needs some improvement. The reviewer highly recommends that the authors do a thorough proof reading of the manuscript.

2-    In the introduction section I believe the authors need to include the most recent advances on soil liquefaction which considers biaxial effects, some of these references are:

El-Shafee O., Abdoun T., Zeghal M. (2016) “Centrifuge modeling and analysis of site liquefaction subjected to biaxial dynamic excitations”, Geotechnique, 67, No. 3, 260-271.

El Shafee O, Abdoun T, Zeghal M (2018) “Physical modeling and analysis of site lique-faction subjected to biaxial dynamic excitations” Innovative Infrastructure Solutions; 3(1):173–185.

Reyes A, Adinata J, Taiebat M (2019) “Impact of bidirectional seismic shearing on the volumetric response of sand deposits” J. Soil Dynamics & Earthquake Engrg, V.125.

Reyes A, Adinata J, Taiebat M (2019) “Liquefaction hazard evaluation under bidirectional seismic shearing: Optimal ground motion intensity measures” Proceedings of VII ICEGE 7th International Conference on Earthquake Geotechnical Engineering, Rome, Italy.

Zeghal M, El-Shafee O, Abdoun T (2018) “Analysis of soil liquefaction using centrifuge tests of a site subjected to biaxial shaking”, J. Soil Dynamics & Earthquake Engrg, V. 114, 229-241.

3-    Tables and figures should be enclosed in the same page with their caption, unless the table is larger than one full page (Table 1, Figure 3, Figure 7, Figure 8, Table 5, Table 7, Figure 13, and Table 9).

4-    The conclusions section summarizes the authors’ observations pretty well. However, the message they want to deliver, or the recommendation based on the former observations is not as clear. Please try to improve that part.

1-    The paper writing has several awkward phrases, and the overall language quality needs some improvement. The reviewer highly recommends that the authors do a thorough proof reading of the manuscript.

Author Response

(The authors gave the same response as above.)

Reviewer 4 Report

1. What kind of similarity is guaranteed in the derivation of similitude coefficient in Table 2? How does the seismic similarity is guaranteed?

2.Some of the units in table 3 are italic and some are not. 

3.The effect of thickness of non-liquefiable soils on the result should be addressed.

Author Response

(The authors gave the same response as above.)

Reviewer 5 Report

This study examines the reliquefaction of the foundation ground of embankment constructed in coastal areas using model vibration experiments in a 1g field. In the experiment, the embankment composed of gravel was constructed on the foundation ground of 50 cm thickness, and the liquefaction of the foundation ground was considered when the excitation was given five times at the same acceleration level. As a result, in the case where the groundwater level is shallow, liquefaction gradually ceases due to the increase in densification of the ground due to the shaking. While in the case where the initial groundwater level is relatively deep, because of the groundwater level rises after several times of shaking, it was obtained the result of liquefying at the third time. In other words, the problem of reliquefaction is regarded as the problem of changes in ground density and groundwater level when subjected to multiple seismic motions.

Based on the results of a limited number of experimental cases, only very obvious conclusions can be drawn. What seems to be a new finding is that experiments have shown that even if the same ground motion does not liquefy in the main shock, it will liquefy in the aftershocks if the material properties and boundary conditions change due to multiple earthquake motions. Although this research is meaningful as one experimental result, it seems that the analysis and consideration of the experimental results are insufficient. For reference, here are the reviewer's comments.

 Reviewer’s comments

1) Lack of reviews on re-liquefaction. Clarify what the problem is and what needs to be clarified.

2) Clarify the experimental conditions, such as intervals between multiple vibrations.

3) Specify the specific evaluation method for ground density.

4) Specify the specific calculation method for the effective vertical confining pressure at the installation position of the pore water pressure gauge. For example, do you take into account the volume change of the sponge at the edge of the shaking table? In addition, changes in the groundwater level and ground density have occurred in the experiment. It is also unknown whether the effective confining pressure was strictly calculated for each excitation.

5) In addition to the above, the groundwater level position prior to each excitation step should also be indicated.

6) It seems that the occurrence or not of liquefaction is defined by whether or not the effective confining pressure ratio exceeds 1, but isn't this definition easy? From the time history of excess pore water pressure, there are also cases where the water pressure ratio exceeds 1 and differences in water pressure amplitude. I would like a deeper consideration.

7) Page 13, First Paragraph:

The notation of the installation position of the excess pore water pressure gauge is expressed in terms of the prototype scale not model scale.

In general, even if the relative density is 70-80%, liquefaction occur if the excitation level is increased according to the authors' definition of liquefaction. The convergence of the relative density to a certain value should be noted only when the same excitation level is applied.

8) In the case-2 experiment, when liquefaction does not occur in the first and second times, the density of the ground does not change greatly, so the change in the groundwater level is also small. In that case, the third condition does not change much either, so the authors' logic leads to the conclusion that nothing changes.

9) In general, relative density is a measure of the density of a specific soil, so use it with that in mind. What can be said about the 60% relative density of silica sand No. 7 does not apply to 60% of another soil.

Author Response

(The authors gave the same response as above.)

Round 2

Reviewer 1 Report

Thanks to the author for the revision. The author has made a response according to my review comments. This article can be accepted for publication.

Author Response

The authors would like to sincerely thank the reviewer for your valuable comments regarding the manuscript.

Reviewer 2 Report

The authors have made careful modification, responding to my concern.

Overall, it is ok.

Author Response

(The authors gave the same response as above.)

Reviewer 5 Report

The authors have responded to the reviewers' comments.

Author Response

(The authors gave the same response as above.)
